# First Insights into the Repertoire of Secretory Lectins in Rotifers

**DOI:** 10.3390/md20020130

**Published:** 2022-02-09

**Authors:** Marco Gerdol

**Affiliations:** Department of Life Sciences, University of Trieste, Via Giorgieri 5, 34128 Trieste, Italy; mgerdol@units.it

**Keywords:** rotifera, pattern recognition receptors, microbe-associated molecular patterns, innate immunity, C-type lectins, C1q domain-containing proteins, galectins

## Abstract

Due to their high biodiversity and adaptation to a mutable and challenging environment, aquatic lophotrochozoan animals are regarded as a virtually unlimited source of bioactive molecules. Among these, lectins, i.e., proteins with remarkable carbohydrate-recognition properties involved in immunity, reproduction, self/nonself recognition and several other biological processes, are particularly attractive targets for biotechnological research. To date, lectin research in the Lophotrochozoa has been restricted to the most widespread phyla, which are the usual targets of comparative immunology studies, such as Mollusca and Annelida. Here we provide the first overview of the repertoire of the secretory lectin-like molecules encoded by the genomes of six target rotifer species: *Brachionus calyciflorus*, *Brachionus plicatilis*, *Proales similis* (class Monogononta), *Adineta ricciae*, *Didymodactylos carnosus* and *Rotaria sordida* (class Bdelloidea). Overall, while rotifer secretory lectins display a high molecular diversity and belong to nine different structural classes, their total number is significantly lower than for other groups of lophotrochozoans, with no evidence of lineage-specific expansion events. Considering the high evolutionary divergence between rotifers and the other major sister phyla, their widespread distribution in aquatic environments and the ease of their collection and rearing in laboratory conditions, these organisms may represent interesting targets for glycobiological studies, which may allow the identification of novel carbohydrate-binding proteins with peculiar biological properties.

## 1. Introduction

Lectin-like molecules play a fundamental role in several physiological processes shared by all animals, including, critically, the discrimination between “self” and “nonself” through the specific recognition of carbohydrate moieties exposed on cellular surfaces. These glycans, when associated with microorganisms, are generally referred to as microbe-associated molecular patterns (MAMPs) or, in the case of potentially pathogenic microbes, pathogen-associated molecular patterns (PAMPs) [1].

The proteins expressed by the host that are involved in carbohydrate recognition are collectively known as pattern recognition receptors (PRRs), which may exert their function at different levels, i.e., in the extracellular environment, at the plasma membrane or within the cell. In the context of immune response, the activity of a heterogeneous group of small secretory PRRs usually leads to the coating of invading microbes. This process may in turn trigger a complex response involving several additional molecular and cellular players which vary widely along the metazoan tree of life. These include, among others, the melanization cascade (typically observed in arthropods and other invertebrates) [2], the production of a large arsenal of antimicrobial peptides [3], the activation of the complement system (well-described in vertebrates and present in a primitive form also in many invertebrates) [4] and the recruitment of specialized phagocytic cells [5]. Furthermore, the ability to recognize MAMPs and to modulate immune responses has been linked with the maintenance of gut microbiome homeostasis [6], as well as the establishment of beneficial bacterial symbioses [7], which are particularly relevant in aquatic environments [8].

Besides their key role in immune recognition, lectins are involved in a number of other physiological processes, to which they contribute thanks to their remarkable ability to recognize glycan moieties with high specificity. For example, some lectins play an important role in reproduction and gamete recognition [9,10], in the clearance of apoptotic cells thanks to the recognition of damage-associated molecular patterns (DAMPs) [11], in larval settlement and metamorphosis [12] and in the recognition of food particles in filter-feeding bivalves [6,13].

Aquatic invertebrates have been a preferred target for lectin identification and purification during the past three decades, as revealed by the fact that many of the best functionally characterized lectins from non-vertebrate metazoans derive from corals, echinoderms and mollusks [14,15,16]. Among the Lophotrochozoa, one of the two clades of spiralian animals together with Ecdysozoa, most glycobiological and immunological studies have been so far focused on species belonging to the phyla Mollusca or Annelida, amenable for research due to their relatively large body size and the ease of sampling and laboratory handling [17]. Other lophotrochozoan phyla have been nearly completely neglected up to now, leaving a remarkable gap of knowledge concerning the main molecular players involved in carbohydrate recognition.

Among these, the phylum Rotifera, which comprises over 2000 described species with a widespread distribution in freshwater environments, but occasionally found also in brackish and saltwater habitats, represents a particularly intriguing unexplored resource for lectin research. Rotifera are classically subdivided between two classes, namely, Monogononta (the most species-rich class) and Bdelloidea, even though phylogenetic evidence suggests that Seisonidea and Acanthocephala also belong to the very same monophyletic group. Bdelloids display a few peculiar features compared with all other lophotrochozoans, such as a remarkable ability to withstand extreme temperatures [18] and ionizing radiations, which is thought to derive from efficient DNA double-strand break repair [19], and obligatory parthenogenetic reproduction, which results from a long-term asexual evolutionary history [20]. Another interesting feature of rotifers lies in their remarkable genetic divergence from the other major lophotrochozoan phyla. Indeed, monogonont and bdelloid rotifer genomes differ greatly, both in terms of size and architecture, which in bdelloids is significantly impacted by the presence of transposable elements [21], massive horizontal gene transfer [22] and signatures of long-term asexual reproduction [23].

Rotifers often belong to cryptic species complexes, which can only be correctly identified through DNA barcoding, and have in most cases a cosmopolitan distribution [24,25]. These organisms, which usually have a very small size (100–1000 μm), constitute a significant fraction of microzooplankton and their biomass can be particularly relevant in certain environments, such as coastal lagoons or shallow, acidified, metal-contaminated lakes [26,27,28]. During the 1970s and 1980s, some rotifer species, such as the eurhyaline *Brachionus plicatilis*, were successfully established as live feeds in marine fish aquaculture, thanks to their fast population growth and ease of intensive culture [29,30,31] (i.e., up to two billion individuals can be obtained in one day per cubic meter of culture [32]). This would undoubtedly represent an interesting opportunity for glycobiology studies, as a sufficient biomass could be readily available for lectin isolation and purification. 

The successful adaptation of rotifers to a challenging environment, where they are potentially exposed to a broad range of microorganisms, suggests that these small animals might have developed carbohydrate-binding strategies similar to those described in other aquatic invertebrates in which multiple biomolecules with high biotechnological potential have been previously identified. Moreover, due to their peculiar features and their high tendency to acquire novel genes by horizontal gene transfer, these small metazoans might be considered as a potential source of isolation for a number of novel lectins with unusual and interesting biological properties.

This work preliminarily explores the repertoire of secretory lectins from six rotifer species belonging to the classes Monogononta and Bdelloidea. The publicly available genomes of these species were screened to look for annotated genes encoding proteins bearing known carbohydrate-binding domains (CRDs). Unlike other lophotrochozoan phyla, in which lectin-like proteins are often encoded by tandemly duplicated paralogous genes displaying high pairwise sequence homology, rotifers do not show evidence of massive gene family expansion events. However, they display a highly diversified arsenal of carbohydrate-binding proteins whose biological properties could be explored and biotechnologically exploited in the near future.

## 2. Results

The screening of six rotifer genomes allowed the identification of a relatively small number of secretory lectin-like molecules compared with other lophotrochozoans, which are often characterized by massive gene family expansion events that involve carbohydrate-binding proteins, as exemplified by the case of C1qDC proteins in bivalves [33,34,35]. Based on available data in the literature, only lectin-like proteins displaying a canonical signal peptide for secretion and which display no significant primary sequence conservation among the different lectin families will be here described; the only exception is represented by galectins, which rely on unconventional secretion.

In the class Monogononta, *Brachionus calyciflorus* was the species in which the highest number of lectins was identified (38), followed by the congeneric species *Brachionus plicatilis* (25) and *Proales similis* (14). In the class Bdelloidea, *Rotaria sordida* and *Adineta ricciae* displayed a similar number of lectins (27 and 22, respectively), whereas the third rotifer species, *Didymodactylos carnosus*, had the lowest number of associated lectin sequences in this study (eight) (Table 1; the full list of gene accession IDs is provided in Appendix A). Based on these observations, it can be estimated that just a very tiny fraction of all protein-coding genes in rotifers (i.e., 0.02–0.15%) encode secretory lectins characterized by the presence of previously described conserved domains. Nevertheless, despite the lack of evident lectin family expansions, the lectin-like proteins identified in all rotifer species displayed a remarkable molecular diversity, as revealed by their classification within nine different families (Table 1): (i) fibrinogen-related domain-containing proteins (FReDs) (Section 2.1); (ii) C-type lectins (Section 2.2); (iii) C1q-domain containing (C1qDC) proteins (Section 2.3); (iv) galectins (Section 2.4); (v) R-type lectins (Section 2.5); (vi) F-type lectins (Section 2.6); (vii) SUEL-type lectins; (viii) H-type lectins; (ix) jacalin-like lectins (Section 2.7).

### 2.1. FReD-Containing Proteins

Fibrinogen-related domain-containing proteins (FReDs) share structural similarity with the C-terminal domain of vertebrate ficolins, i.e., *N*-acetylglucosamine (GlcNAc)-specific carbohydrate-binding proteins, which play a key role in the lectin pathway of the complement system [36,37]. The fibrinogen C-terminal domain is associated with a number of metazoan lectins with widespread taxonomic distribution, from cnidarians to vertebrates, which hold remarkable glycan-binding properties and often play an important role in the context of immune recognition, as revealed by several studies carried out in Mollusca [38,39,40].

A subgroup of FReDs named fibrinogen-related proteins (FREPs), which combine one or two N-terminal immunoglobulin domains with a single C-terminal fibrinogen domain, have been implicated in the resistance of snails to trematode infections [41,42]. Comparative immunogenomics studies have previously revealed that bona fide FREPs [43], as well as GREPs and CREPs (i.e., FReDs associated with galectin and CTL domains, respectively [44]), are restricted to the gastropod subclass Heterobranchia. Nevertheless, other mollusks display a high number of proteins with a simpler architecture, comprising a signal peptide and the fibrinogen-like domain, often paired with a coiled-coil region of variable length, which may allow their oligomerization, in a similar fashion to collagen in vertebrate ficolins [45]. Single-domain FReDs, which retain significant glycan-binding properties in the Lophotrochozoa [46], underwent a significant expansion in bivalves, where they are often found with hundreds of paralogous gene copies [47] encoding inducible proteins with marked bacteria-agglutinating properties [48,49]. Similar expansions have certainly occurred in other lophotrochozoan phyla, such as brachiopods, even though the functional implications of these events are presently unclear [50].

All the rotifer species analyzed in this study had FReD genes in varying numbers, ranging from one (in *D. carnosus*) to six (in *R. sordida*, *B. plicatilis* and *A. ricciae*) (Table 1). The encoded proteins from bdelloid and monogonont rotifers displayed different architectures: while all FReDs shared a single peptide and displayed a fibrinogen-like domain in a C-terminal position, they were characterized by the presence of an N-terminal region of variable length (Figure 1). This region was markedly shorter in bdelloid FReDs, which usually displayed a relatively high (55–50%) primary sequence identity with horseshoe crab tachylectins [39], and much longer in monogonont FReDs, which, on the other hand, had a lower homology (i.e., 25–35%) with tachylectins. In all rotifer FReDs, this region lacked detectable conserved domains and structural homologies but displayed a low level of complexity and the occasional presence of threonine-rich amino acid stretches.

### 2.2. C-Type Lectins

C-type lectins (CTLs) are one of the largest and most studied families of lectins in lophotrochozoans, with several dozen proteins having been functionally characterized in mollusks and segmented worms [51,52,53]. Their characterizing CRD, which displays a broad calcium-dependent binding specificity, is often found in large multidomain membrane-bound proteins which may or may not have a lectin function [54,55]. Their remarkable structural diversity has led to the development of a complex classification system, which has been subjected to multiple updates over the years [54,56,57]. Since such a classification still appears to be strongly biased towards vertebrates, it is not fully adequate to describe the variegate domain combinations found in animal CTLs.

Compared with their membrane-bound counterparts, secretory CTLs usually display a simpler structure, which comprises a signal peptide, followed by either one or two tandemly repeated CRDs. In addition, the N-terminal region may also include coiled-coil or collagen repeats with effector functions [58,59]. Besides having a role in MAMP recognition, the CTLs of invertebrate metazoans can regulate different aspects of the innate immune response, including microbial opsonization, the activation of the prophenoloxidase-mediated melanization cascade and possibly also the activation of the complement system, mirroring the role of the mannan-binding lectin in the lectin pathway of the vertebrate complement system [52,60,61].

As far as the Lophotrochozoa are concerned, multiple studies have previously revealed that CTLs belong to highly expanded gene families in Mollusca [33,62], Annelida and Brachiopoda [50]. The investigations carried out here in Rotifera revealed a highly variable number of CTLs among species. While CTLs represented the largest group of secretory lectins in the genus *Brachionus* (i.e., 25 in *B. calyciflorus* and 17 in *B. plicatilis*), only a few proteins of this type (i.e., one to four) could be identified in the four other species (Table 1). Most of the proteins identified in *Brachionus* spp. had a single CRD (Figure 1), which often followed a relatively long (i.e., ~100 amino acids) N-terminal region with no recognizable conserved domains. In addition, both *Brachionus* species displayed a few CTLs with two consecutive CRDs, whose architecture resembled those of insect immulectins [63]. Another type of domain combination included the presence of an epidermal growth factor (EGF)-like domain, placed immediately before the CRD. EGF domains are often found in association with certain large vertebrate CTLs found in the extracellular matrix or bound to the cell membrane, such as selectins and lecticans. However, the combination of a single EGF domain and the CTL CRD has never been described before in the Lophotrochozoa. The third analyzed monogonont rotifer species, *P. similis*, only displayed three CTL genes: two encoded short, single-domain lectins, whereas the third one had an additional EGF-like domain, as previously described in *Brachionus* spp. (Figure 1).

The three bdelloid species had a smaller number of genes encoding secretory CTLs: three were identified in *A. ricciae*, two in *R. sordida* and a single one in *D. carnosus*. Two CTLs from *R. sordida* and one from *A. ricciae* were short single-domain CTLs. *D. carnosus* and *A. ricciae* shared the presence of an orthologous sequence with two recognizable CRDs located at the N-terminal end, followed by a long region with no detectable primary sequence or structural homologies. The third CTL identified in *A. ricciae* showed an unusually long N-terminal low-complexity region, highly enriched in threonine and serine residues, followed by a C-terminal CRD (Figure 1).

In general, rotifer CTLs only showed a poor primary sequence homology (i.e., 20–30%) with functionally characterized molluscan CTLs, which prevented the ascertainment of clear orthology relationships. It is worth mentioning that a single protein belonging to the CTL family had been previously described and functionally characterized in Rotifera. Nevertheless, the sequences orthologous with this protein, which serves as the mate recognition pheromone in the male individuals of *Brachionus manjavacas* [64], are not reported in the present study due to the presence of a transmembrane domain.

### 2.3. C1q Domain-Containing Proteins

C1q domain-containing (C1qDC) proteins belong to a widespread family of highly versatile globular proteins with remarkable binding properties [65,66]. Besides their well-characterized involvement in the vertebrate complement system, C1qDC proteins carry out important functions in other biological processes which have only recently started to be unveiled [67]. For example, thanks to the carbohydrate-binding properties demonstrated in several metazoan phyla [68,69], C1qDC proteins should be regarded as PRRs involved in immune recognition. This role has been investigated in detail in Mollusca [70,71], where C1qDC proteins are associated with massive gene family expansions [34,35,72]. In bivalves, such expansions involve C1qDC proteins that either have a very simple architecture (signal peptide + C1q domain) or contain an additional N-terminal coiled-coil region. Moreover, in some gastropod species, such as *Littorina littorea*, the C1q domain is combined with one or two immunoglobulin-like domains, originating a small class of proteins known as QREPs, which are upregulated in response to *Himasthla elongata* infections [73]. 

Unlike bivalves but similar to other lophotrochozoan phyla, such as annelids and brachiopods [50,73], rotifers only display a very few secretory C1qDC proteins (Table 1). In detail, a single orthologous C1qDC gene could be identified in the three monogonont rotifer species, whereas the three bdelloid rotifers had a variable number of C1qDC genes, ranging from two (in *D. carnosus*) to six (in *R. sordida*), with evidence of a few nearly identical paralogs (further supported by phylogenetic evidence; see below). In all cases, rotifer C1qDC proteins were relatively short (<350 aa) and displayed a single C-terminal C1q domain (Figure 1). All proteins had a short (~30 aa long) collagen-like region placed immediately before the start of the C1q domain, which was characterized by the presence of nine highly conserved glycine residues (Figure 2A). This domain organization denotes the typical structure of C1q-like proteins, which represent the most common type of C1qDC proteins in vertebrates [50]. C1q-like proteins are present (but rare) in the lophotrochozoan species characterized by C1qDC gene family expansions, in which collagen repeats are usually replaced by coiled-coil regions [34].

From a phylogenetic point of view, the C1qDC proteins of rotifers were subdivided into three distinct groups (Figure 2B): the first included the C1qDC proteins from Monogononta, which were clustered with high support (posterior probability = 0.99) with a few C1q-like proteins previously identified in other lophotrochozoans and hypothesized to play a key role in the proto-complement system [73]. The C1qDC proteins from bdelloid rotifers were clustered in two groups: the first one, which included a few highly similar paralogous genes in each species (two in *D. carnosus* and *A. ricciae*, four in *R. sordida*), comprised proteins with high sequence homology relative to the group of C1qDC proteins from Monogononta and other lophotrochozoans described above. These proteins displayed, as a peculiar feature, an N-terminal low complexity Ser- and Gln-rich region. The second group of C1qDC proteins from bdelloids only comprised sequences from *A. ricciae* and *R. sordida*, which displayed a high divergence with all the other sequences mentioned above and may therefore represent bdelloid innovations. 

### 2.4. Galectins

Galectins are taxonomically widespread and structurally well-conserved β-galactosyl-binding lectins which carry out a multitude of different functions, including cell adhesion, cellular homeostasis and self/non-self and microbial recognition [74]. Based on their structural organization, lophotrochozoan galectins can generally be considered as belonging to the “tandem-repeat” subtype and contain either two or four CRDs [75,76,77,78], with rare occurrences of galectins with three CRDs [50]. Although phylogenetic analyses have previously revealed a monophyletic origin for all molluscan galectins [79], it is presently unclear whether this consideration also applies to the galectins from other lophotrochozoan phyla.

This investigation allowed the identification of galectin genes in all the six analyzed rotifer genomes, even though their number significantly varied among species. While all Monogononta only had a single galectin, bdelloid genomes encoded multiple galectin genes, ranging from two (*D. carnosus*) to eight (*R. sordida*) (Table 1). All rotifer galectins displayed two tandemly repeated CRDs, separated by a connecting region of variable length (Figure 1). No galectins with four CRDs could be identified, confirming the previous observation that, within the Lophotrochozoa, this subtype is restricted to brachiopods, phoronids and annelids [50]. Primary sequence homology with other members of the galectin family from non-rotifer lophotrochozoans was generally in the range of 30–35%. Consistently with previous observations in other metazoans, the encoded proteins lacked a canonical signal peptide and might therefore use an alternative secretion route [80]. 

### 2.5. Ricin β-Trefoil Lectins

The R-type lectin (RTL) domain, originally described in the plant toxin ricin, is found in a number of metazoan multidomain proteins with different functions, including hydrolases, glycosyltransferases and membrane-bound receptors [81]. Nevertheless, smaller proteins with no additional domains can serve as lectins in the extracellular environment, playing a role in PAMP recognition. A number of secretory R-type lectins with different glycan-binding properties, containing either one or two consecutive CRDs, have been previously isolated in annelids [82,83,84] and mollusks [85]. A second family of lectins, named mytilectins, which share the same β-trefoil three-dimensional structure but do not conform with the canonical R-type lectin primary sequence signature, show a discontinuous distribution among the Lophotrochozoa and have only been described so far in a few bivalve mollusks and brachiopods [50,86,87].

While rotifer genomes encoded several proteins with R-type lectin domains, in most cases these were associated with other domains known to exert catalytic activities (e.g., glycosylases, hydrolases, etc.) or with transmembrane domains. Strong evidence in support of the existence of secretory RTLs could be collected only for two out of the six rotifer species analyzed in this study, i.e., *R. sordida*, among bdelloids, and *B. calyciflorus*, among Monogononta (Table 1).

In detail, the three secretory RTLs identified in *B. calyciflorus* displayed an unusual domain architecture, never before reported in other metazoans. Indeed, these proteins showed the presence of two consecutive VOMI (vitelline membrane outer layer protein I) domains, followed by a C-terminal ricin-like CRD (Figure 1). Although the VOMI domain is typically found in proteins found in the outer layer of the egg vitelline membrane [88], it shares a β-prism fold that has been previously identified in other carbohydrate-binding proteins, including jacalins, a class of plant-specific lectins [89,90,91], as well as in the *B. thuringiensis* delta endotoxin [92]. Due to the simultaneous presence of these two structurally different CRDs, which clearly presents an interesting path for exploration in glycobiological studies, we defined these unusual proteins as BPBT (β-prism, β-trefoil) lectins. Two BTBP lectins, orthologous to those found in *B. calyciflorus* but lacking a signal peptide (possibly due to an incorrect annotation), were also found in the congeneric species *B. plicatilis*, but not in the other species, suggesting that this domain combination may be exclusively present in *Brachionus* spp.

On the other hand, the single secretory RTL found in *R. sordida* was unrelated to BPBT lectins, since this protein was relatively short (i.e., 200 amino acids) and included a chitin-binding domain in an N-terminal position [93] (Figure 1). This domain is shared by several chitinases and other smaller chitin-binding proteins, which include some with demonstrated effector activity in the context of invertebrate innate immunity, such as horseshoe crab tachycytin [94] and mussel mytichitin [95], and others with presumed lectin-like functions [96].

No sequence orthologous to brachiopod and molluscan mytilectins could be found in rotifers, confirming the discontinuous taxonomic distribution of these β-trefoil lectins in the Lophotrochozoa.

### 2.6. F-Type Lectins

F-type lectins are characterized by the presence of a β-barrel jellyroll fold which allows fucose recognition [97] and which is also found in the C-terminal domain of coagulation factors 5/8. Despite being associated with relatively short secretory proteins with a lectin function, the typical CRD of FTLs is often found in large multidomain proteins with different catalytic activities [98]. The frequent combination of this domain with several other non-lectin domains mirrors the previously mentioned functional plasticity of the CRDs of CTLs and RTLs. Previous studies have reported that FTLs underwent expansion in some gastropod species [77], and some functional evidence collected in bivalves has linked these proteins to bacterial recognition [99], in addition to the well-established role of the FTL domain-containing proteins bindins in gamete recognition [100]. This observation is consistent with the detection of the FTL domain in a relatively high number of rotifer proteins, only a few of which were characterized by the presence of a signal peptide or displayed a domain organization consistent with a lectin function (Table 1).

Two different types of secretory F-type lectin sequences were detected in rotifers. The first type, present as a single-copy gene in the three monogonont species but missing in the three bdelloids, was a protein displaying a low-complexity threonine- and serine-rich N-terminal region, followed by a single CRD lacking any significant primary sequence homology with known FTLs but showing high predicted structural similarity with human coagulation factors [101] and discoidins [102] (Figure 1). The second type, shared by all rotifer species (even though *B. plicatilis* only displayed a protein lacking the signal peptide, likely due to incorrect annotation), displayed three consecutive FTL CRDs (Figure 1). While the first and the second ones were well recognizable, the third one did not conform with the canonical F-type lectin signature. These triple-CRD FTLs displayed a relatively high (i.e., 40%) sequence identity with several proteins encoded by the genomes of other lophotrochozoans, including mollusks and annelids, suggesting a high degree of evolutionary conservation.

### 2.7. Other Types of Lectins

In sea urchins, a group of lectins, characterized by the presence of a D-galactoside/L-rhamnose-binding SUEL (acronym for sea urchin egg lectin) domain, carry out egg-protecting functions [103,104]. While this type of lectins is also found in lophotrochozoan genomes, they have been better functionally characterized in deuterostome invertebrates [105,106,107]. To date, their role in lophotrochozoans remains elusive, even though a bivalve SUEL-type lectin was shown to promote the agglutination of Gram-negative bacteria through LPS binding [108]. The SUEL domain was present in some large multidomain membrane-associated proteins of bdelloids but not in secretory proteins. On the other hand, the three monogonont rotifer genomes encoded a few short SUEL-type lectins (Table 1), which lacked accessory conserved domains (Figure 1) and did no bear any detectable primary sequence homology with other metazoan sequences with known functions. With the exception of two sequences detected in *P. similis*, these proteins displayed high pairwise primary sequence homology and clearly belonged to a monophyletic family.

H-type lectins (HTLs) represent a poorly functionally characterized family of *N*-acetylgalactosamine-binding lectins, which are believed to carry out a defensive role against bacterial infections in fertilized snail eggs [109,110]. Although very little information is available about the involvement of HTLs in lophotrochozoan immunity, comparative genomics analyses indicate that they do not belong to expanded gene families, neither in Mollusca [111,112] nor in Brachiopoda [50]. Nevertheless, transcriptome scans carried out in gastropod mollusks revealed the presence of a novel domain combination between immunoglobulin-like domains and HTL domains in the so-called HREPs [73]. The analysis of rotifer genomes revealed the presence of secretory H-type lectins in just two out of the three bdelloid species (i.e., *A ricciae* and *R. sordida*). On the other hand, no HTL was identified in Monogononta (Table 1). These proteins had a similar simple architecture, with a single CRD, placed immediately after a well-recognizable signal peptide (Figure 1). Rotifer HTLs were encoded by open reading frames with a relatively small size (i.e., 120 codons) and displayed poor sequence homology with other known sequences (i.e., less than 40% primary sequence homology vs. *L. anatina*). This may suggest that the apparent lack of secretory HTLs in four out of six target species derives from missing gene models that could not be included in the annotation tracks of the respective genomes due to poor supporting evidence.

Section 2.5 reports the presence of BPBT lectins, which bear a jacalin-like β-prism structural domain in combination with the RTL CRD, in *Brachionus* spp. The screening for additional proteins bearing a canonical jacalin domain led to the identification of a single protein in *D. carnosus* with no orthologs in other rotifer species. This lectin, which displayed a well recognizable signal peptide for secretion, lacked significant primary sequence homology with other previously characterized proteins, but displayed a highly significant structural match with a number of jacalin-like lectins from plants and with a few metazoan proteins. These included, as the only lophotrochozoan representative, the PPL3 lectin from the bivalve mollusk *Pteria penguin*, which is involved in shell mineralization [113]. Other relevant metazoan proteins which display the same structural fold are the human pancreatic secretory protein ZG16b, important for the condensation of pancreatic enzymes [114], the WGA16 protein from boar sperm [115] and the zebrafish pore-forming protein Dln1 [116]. 

No gene encoding proteins homologous to the egg-protecting lectins from *Aplysia dactylomela*, characterized by the presence of the orifera lack lectins homologous to her metazoan sequences with known function.ked significant similarity withology DUF3011 domain [117], could be detected in Rotifera. Likewise, no apextrin-like proteins were detected in any of the studied species. Proteins carrying an apextrin C-terminal domain (ApeC) have been previously shown to mediate bacterial recognition in amphioxus [118]. While they are also present in bivalve mollusks and brachiopods [50,62], this study rules out their possible involvement in immune recognition in rotifers.

## 3. Discussion

Genome- or transcriptome-wide screening approaches have previously been successfully used to identify lectin-like proteins in several different eukaryote species, both in the plant and animal kingdoms [77,96,112,119,120,121,122], and can be effectively used as a preliminary step to investigate the repertoire of lectin-like molecules present in non-model species. This approach might be particularly intriguing for understudied animal phyla, which, despite the lack of previous glycobiological investigations, might be endowed with a particularly rich repertoire of lectin-like molecules as the result of their adaptation to a challenging environment. Rotifers, like other aquatic organisms, are potentially highly exposed to ubiquitous waterborne microorganisms, which may in some cases be pathogenic. Considering the well-described complex innate immune systems of other lophotrochozoans, such as mollusks and annelids, rotifers appear to be good candidates for the bioinformatics-assisted discovery of carbohydrate-binding proteins involved in MAMP recognition. No information is presently available concerning the glycans expressed by rotifer tissues and only a single membrane-bound C-type lectin has been described in *B. manjavacas* [66] prior to this work. Hence, this represents the first investigation of this type carried out in this relatively large and widespread but poorly studied phylum of small aquatic animals.

Although this approach obviously suffers from some limitations, which will be described in detail below, it has allowed: (i) the identification of the presence of secretory lectin-like molecules belonging to at least nine different families, characterized by distinct structural folds (Figure 3), in rotifers; (ii) the highlighting of significant differences in terms of distribution and domain organization between the two major classes of rotifers, as well as among species; and (iii) the ruling out of the possibility that known lectin-encoding gene families underwent significant expansion in Rotifera, marking a clear difference with other lophotrochozoan phyla.

Even though this in silico screening approach allowed the identification of several proteins that are likely to be secreted to the extracellular environment and have significant carbohydrate-binding properties, the list of the putative rotifer lectin-like proteins here provided (detailed in Appendix A) should be considered as strictly preliminary. The glycan-binding properties of the identified proteins, as well as their possible involvement in MAMP recognition, should be validated through functional data collected with classical biochemical, glycobiological and immunological approaches.

Some limitations of this genome-wide bioinformatics screening approach reside in the fact that the correct identification of lectin-like proteins depends on the accuracy of gene annotations. While all the genomes analyzed in this work had a high quality, both in terms of assembly metrics and in terms of BUSCO completeness [123], a few chimeric gene models, as well as models of ORFs which were clearly subjected to 5’ or 3’ truncation compared with other full-length orthologs and paralogs, were occasionally observed. For the sake of consistency, these gene models were disregarded, even though the presence of incomplete gene models in a given species was reported, whenever relevant, in Table 1. In addition, some lectin-like proteins identified in this work were rather short, with an ORF barely exceeding 100 codons, and lacked at the same time significant primary sequence homology with other known sequences deposited in public databases. These factors might have negatively impacted the annotation of other orthologous and paralogous genes, which may therefore be apparently missing in some of the target genomes, as discussed in Section 2.7. Hence, the completeness of our report may suffer from these uncertainties, and the number of lectins reported in Table 1 should be considered as inherently subjected to slight underestimates. 

Another possible limitation of this work was the impossibility of proceeding with a reliable in silico screening of candidate lectin molecules characterized by the presence of domains which are not primarily or exclusively linked with a carbohydrate-binding function. This was the case, for example, for I-type lectins (also known as siglecs), which share an immunoglobulin-like fold with a very high number of other proteins involved in a very broad range of functions [124] and which have been previously identified in some lophotrochozoans [125]. Similarly, some chitin-binding lectins, mostly from plants [126], are characterized by the presence of a chitinase-like domain that includes a few key mutations that ablate its catalytic function. However, since these two lectin families are either membrane-bound (siglecs) or uncommon in metazoans (chitinase-like lectins), their exclusion from the set of domains included in in silico searches was unlikely to have an impact on the identification of secretory lectins in rotifers.

Finally, it needs to be stressed that rotifer genomes are extremely gene-rich and encode several thousand proteins which lack any significant primary sequence homology with other sequences deposited in public databases and which have no detectable conserved domain. We cannot exclude that some as yet uncharacterized protein families may carry out important carbohydrate-binding functions in these animals. Nevertheless, the combination of classical biochemical and glycobiological approaches with bioinformatics approaches should enable the identification of the full-length sequence of the lectins isolated from rotifers starting from small peptide fragments, as previously carried out on several occasions with other aquatic invertebrates [68,85,127].

In summary, this work allowed the confirmation of the potential interest of rotifers as future targets for glycobiological studies focused on the identification of novel lectins. These, based on the significant diversity of the associated structural folding, might be endowed with different carbohydrate-binding properties, which may support the development of new biotechnological tools, such as lectin-based biosensors with potential applications in cancer research. Besides the interest that such molecules might have in terms of biotechnological applications, another aspect that remains to be clarified is whether these rotifer secretory lectins carry out biological functions similar to those previously described in other lophotrochozoan phyla.

## 4. Materials and Methods

### 4.1. Identification of Lectin-Like Molecules

Six rotifer species with a publicly available fully sequenced genome and an associated gene annotation track were selected (Table 2). *Didymodactylos carnosus* Milne 1916 [21], *Rotaria sordida* Western, 1893 [21] and *Adineta ricciae* Segers & Shiel, 2005 [21] were selected for the class Bdelloidea; *Brachionus plicatilis* Müller, 1786 [128], *Brachionus calyciflorus* Pallas, 1776 [129] and *Proales similis* de Beauchamp, 1907 [130] were selected for the class Monogononta.

The proteome of each of the six target species was screened and a search was made for secretory proteins, i.e., those including either a highly supported canonical signal peptide, detected with SignalP v.5.0 [131], or, in the case of galectins, which are known to use a non-canonical secretion signal, with SecretomeP v.2.0 [132]. At the same time, candidate proteins needed to lack transmembrane regions, which were detected with TMHMM v.2.0 [133]. Signal peptide and transmembrane region predictions were further checked with Phobius v.1.01 [134]. Putative lectins were identified based on the presence of the following PFAM conserved domains, detected with HMMER [135] based on default threshold e-values: fibrinogen beta and gamma chains, C-terminal globular domain (PF00147), C-type lectin domain (PF00059), C1q domain (PF00386), galactose-binding lectin domain (PF02140), ricin-type beta-trefoil domain (PF00652 and PF14200), F-type lectin/discoidin domain (PF00754), galactoside-binding lectin domain (PF00337), H-type lectin domain (PF09458), jacalin-like lectin domain (PF01419), DUF3011 (PF11218) and the C-terminal domain of apextrin (PF16977).

The presence of other conserved domains was checked with InterProScan v.5 [136] and the possible presence of conserved structural folds in regions lacking conserved domains was investigated with HHpred [137]. To avoid the inclusion of proteins carrying lectin-like domains but likely involved in non-lectin functions, sequences displaying conserved domains with known catalytic action (e.g., glucanases, hydrolases, kinases, peptidases, etc.) were disregarded. Protein sequences deriving from gene models annotated as “partial”, as well as those that displayed obviously truncated characterizing domains and which might therefore derive either from pseudogenes or from mis-annotations, were disregarded.

### 4.2. Bayesian Phylogenetic Inference

All rotifer C1qDC proteins were included in a multiple sequence alignment (MSA), prepared with MUSCLE [138], together with three selected lophotrochozoan C1q-like sequences (i.e., XP_013399541.1 from *Lingula anatina*, QBA18422.1 from *Testudinalia testudinalis* and XP_033760315.1 from *Pecten maximus*). The human C1qA (NP_057075.1), C1qB (NP_000482.3) and C1qc (NP_758957.2) chains were also added to the alignment for tree-rooting purposes. The MSA was refined with Gblocks v.0.91b [139] to remove unalignable, poorly informative regions. Bayesian phylogenetic analysis was carried out with MrBayes v.3.2.7a [140], running two parallel MCMC analyses for 500,000 generations each, sampling one tree every 1,000 generations. The analysis was run under an LG+G+I model of molecular evolution, which was estimated to be the best fitting for this dataset, with ModelTest-NG [141]. Run convergence was checked with Tracer v.1.7.1 [142] by determining that all estimated parameters reached an effective sample size value higher than 100.

## Figures and Tables

**Figure 1 marinedrugs-20-00130-f001:**
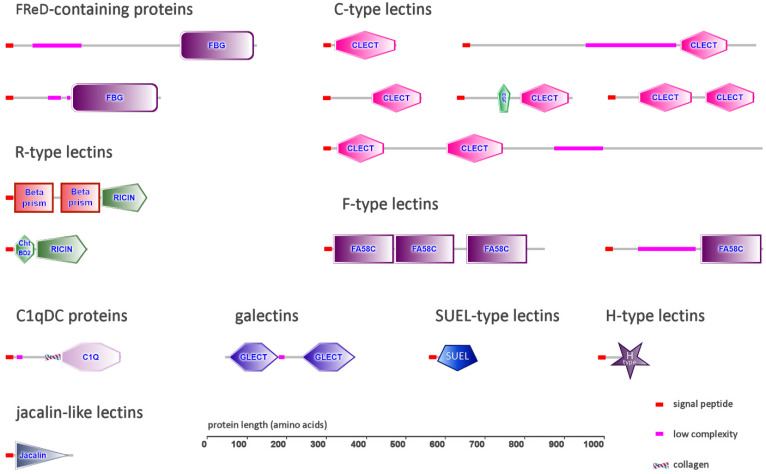
Schematic representation of the main type of secretory lectin-like molecules identified in Rotifera. FBG: Fibrinogen C-terminal domain; GLECT: galectin domain; SUEL: D-galactoside/L-rhamnose-binding SUEL lectin domain; CLECT: C-type lectin domain; FA58C: coagulation factor 5/8 C-terminal domain; EGF: epidermal growth factor domain; Cht BD2: chitin binding domain.

**Figure 2 marinedrugs-20-00130-f002:**
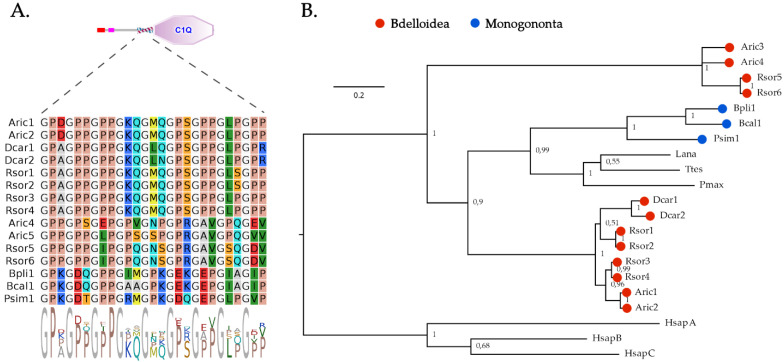
(**A**) Schematic structure of the C1qDC proteins identified in rotifers, with a zoom on the collagen region. (**B**) Bayesian phylogeny of C1qDC proteins from rotifers, obtained with 500,000 generations of an MCMC analysis, run under an LG+I+G model of molecular evolution. The numbers shown close to each node represent posterior probability support values. Aric: *A. ricciae*; Rsor: *R. sordida*; Psim: *P. similis*; Dcar: *D. carnosus*; Bcal: *B. calyciflorus*; Bpli: *B. plicatilis*; Lana: *Lingula anatina*; Pmax: *Pecten maxiumus*; Ttes: *Testudinalia testudinalis*; Hsap: *Homo sapiens*. Human sequences were used as an outgroup for tree-rooting purposes; A, B and C indicate the human C1qA, C1qB and C1qC chains.

**Figure 3 marinedrugs-20-00130-f003:**
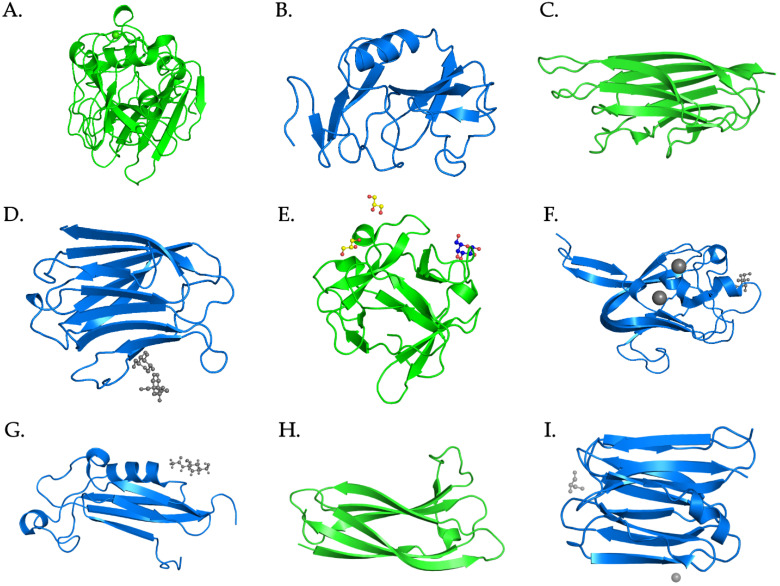
Three-dimensional folding of the carbohydrate-recognition domains of representative members of the nine lectin families identified in Rotifera. (**A**) C-type lectin domain, pdb entry: 1b6e; (**B**) fibrinogen C-terminal domain, pdb entry: 1fib; (**C**) C1q domain, pdb entry: 4ous; (**D**) galectin domain, pdb entry: 1a3k; (**E**) R-type lectin β-trefoil domain, pdb entry: 4iyb; (**F**) F-type lectin domain, pdb entry: 1k12; (**G**) SUEL-type lectin domain, pdb entry: 2jx9; (**H**) H-type lectin domain, pdb entry: 2ces; (**I**) jacalin β-prism domain, pdb entry: 3apa. The figures are reproduced courtesy of PDBe (https://www.ebi.ac.uk/pdbe/, accessed on 20 January 2022).

**Table 1 marinedrugs-20-00130-t001:** Number of secretory lectins identified in the six rotifer species analyzed in this study. The full list of gene accession IDs is provided in Appendix A.

	Bdelloidea			Monogononta		
	*Adineta ricciae*	*Rotaria sordida*	*Didymodactylos carnosus*	*Proales similis*	*Brachionus calyciflorus*	*Brachionus plicatilis*
FReDs	6	6	1	3	2	6
C-type lectins	3	2	1	3	25	17
C1qDC proteins	4	6	2	1	1	1
Galectins	4	8	2	1	1	1
R-type lectins	0	1	0	0	3	0 ^a^
F-type lectins	3	3	1	2	3	0 ^b^
SUEL-type lectins	0	0	0	4	3	0 ^c^
H-type lectins	2	1	0	0	0	0
Jacalin-like lectins	0	0	1	0	0	0
Apextrins	0	0	0	0	0	0
DUF3011 lectins	0	0	0	0	0	0

^a^ Two partial BPBT lectins (see Section 2.5), lacking a signal peptide, likely due to incorrect annotation, were detected. ^b^ A single FTL with three CRDs, lacking a signal peptide, likely due to incorrect annotation, was detected. ^c^ Two short single-domain SUEL-type lectins, encoded by two paralogous genes, were likely incorrectly fused in a single gene model.

**Table 2 marinedrugs-20-00130-t002:** List of the six rotifer species analyzed in this study, with genome size and number of annotated protein-coding genes.

Species Name	Class	Genome Size (Mb)	Protein-Coding Genes
*Adineta ricciae*	Bdelloidea	173	49,015
*Rotaria sordida*	Bdelloidea	361	61,901
*Didymodactylos carnosus*	Bdelloidea	356	46,863
*Proales similis*	Monogononta	33	10,785
*Brachionus calyciflorus*	Monogononta	30	24,328
*Brachionus plicatilis*	Monogononta	107	52,502

## Data Availability

All the genomic data analyzed in this study are publicly available as supplementary materials of the original publications, referenced in the materials and methods section.

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
