# Peer review of "First Insights into the Repertoire of Secretory Lectins in Rotifers"

_marinedrugs, 2022, doi:10.3390/md20020130_

Round 1

Reviewer 1 Report

 Gerdol reports the bio-informatic analysis and characterisation of lectin-like proteins in several rotifer species. This is a well-written article and is very comprehensive to an extent that the introduction could read like a review article, which is not a bad comment. The methods are appropriate and I would recommend acceptance as it is.

Author Response

I would like to thank the reviewer for his/her positive assessment of this work.

Reviewer 2 Report

The manuscript by Gerdol reports the first study aimed at a describing the repertoire of the secretory lectin-like molecules encoded by the genomes of six target rotifer species. This manuscript is excellent, being well written and complete, the conclusions are justified by the results and are well discussed in terms of context and limitations. The subject deserves to be published in Marine Drugs. Here are some comments and minor corrections to be made to the text and content:

  1. Signal peptides: since candidate secreted lectins were identified based on the presence of a signal peptide and the lack of transmembrane regions, it would be appropriate to elaborate more in the results section on the different and/or main signal peptides that were found. Maybe a table summarizing the signal peptides found would be useful.

  1. Massive gene family expansion events (such as mentioned at lines 103-104): for the readers that are unfamiliar with such, please elaborate more on what it could have looked like, how to recognize those, since no instance were identified in this study.

  1. While the potential of rotifers was well described in the introduction (namely lines 86-99), it would be useful also to elaborate more on their potential in the concluding statement / discussion. Please also do elaborate more in the discussion on the potential contributions and novelty of this study.

  1. Out of curiosity, what is known regarding the glycans expressed by rotifer species and how could this hint at the discovery of potentially new lectins? Could also some of these novel glycans have potential as biotherapeutics and/or tools for glycobiology?

  1. Keywords: Microbe-Associated Molecular Patterns
  2. Line 52: Damage-Associated Molecular Patterns
  3. Line 86: ‘80s
  4. Table 1: It would look better if the species name were written without cuts (sor- dida), like Adineta ricciae and Brachionus plicatilis are. Also, if it would be possible to add horizontal lines to clearly distinguish visually which species belong to Bdelloidea and to Monogononta.
  5. Line 138: according to conventions in writing carbohydrates, the “N” of N-acetylglucosamine needs to be in italics.
  6. Line 173: according to conventions in writing carbohydrates, “D” and “L” needs to be smaller. Carbohydrate Research (ISSN 0008-6215) recommends writing these characters as 2 font size smaller than the text.
  7. Line 369: here, SUELs are described as “rhamnose-binding lectins”, yet on line 173, they are also described as “D-gal/L-rha binding SUEL lectin domain”. Would it be possible to adjust this inconstistency, maybe by adding some more information in the results section?
  8. Line 382: same as comment 9.

Author Response

The manuscript by Gerdol reports the first study aimed at a describing the repertoire of the secretory lectin-like molecules encoded by the genomes of six target rotifer species. This manuscript is excellent, being well written and complete, the conclusions are justified by the results and are well discussed in terms of context and limitations. The subject deserves to be published in Marine Drugs. Here are some comments and minor corrections to be made to the text and content:

  1. Signal peptides: since candidate secreted lectins were identified based on the presence of a signal peptide and the lack of transmembrane regions, it would be appropriate to elaborate more in the results section on the different and/or main signal peptides that were found. Maybe a table summarizing the signal peptides found would be useful.

Signal peptides are relatively short (20-25 amino acids) sequence stretches which share a similar amino acid composition and an over-representation of hydrophobic residues, but usually display little primary sequence conservation, unless they share the same evolutionary origins. In the case of rotifer lectins, the different families showed, as expected, unrelated signal peptides. Within the same lectin families, some conservation was observed only among the sequences encoded by paralogous or orthologous genes.

The following sentence was added at the beginning of the results and discussions section to simplify the interpretation from the readers: “Based on available literature data, only lectin-like proteins displaying a canonical sig-nal peptide for secretion, which displayed no significant primary sequence conserva-tion among the different lectin families, will be here described; the only exception is represented by galectins, which rely on unconventional secretion.”

  1. Massive gene family expansion events (such as mentioned at lines 103-104): for the readers that are unfamiliar with such, please elaborate more on what it could have looked like, how to recognize those, since no instance were identified in this study.

Thank you for this suggestion. The sentence was modified as follows: “Unlike other lophotrochozoan phyla, where lectin-like proteins are often encoded by tandemly duplicated paralogous genes displaying high pairwise sequence homology, rofifers do not show evidence of massive gene family expansion events.”

  1. While the potential of rotifers was well described in the introduction (namely lines 86-99), it would be useful also to elaborate more on their potential in the concluding statement / discussion. Please also do elaborate more in the discussion on the potential contributions and novelty of this study.

 This aspect was expanded in the discussion section. The following text was added:

“This approach might be particularly intriguing for understudied animal phyla, which despite the lack of previous glycobiological investigations, might be endowed with a particularly rich repertoire of lectin-like molecules, as the result of their adaptation to a challenging environment. Rotifers, like other aquatic organisms, are potentially highly exposed to ubiquitous waterborne microorganisms, which may be in some cases pathogenic. Considering the well-described complex innate immune systems of other lophotrochozoans, such as mollusks and annelids, rotifers appear to be good candi-dates for the bioinformatics-assisted discovery of carbohydrate-binding proteins in-volved in MAMP recognition. No information is presently available concerning the glycans expressed by rotifer tissues and only a single membrane-bound C-type lectin had been described in B. manjavacas [66] prior to this work. Hence, this represents the first investigation of this type carried out in this relatively large and widespread, but poorly studied phylum of small aquatic animals.”

And

“These, based on the significant diversity of the associated structural folding, might be endowed with different carbohydrate-binding properties, which may support the development of new biotechnological tools, such as lectin-based biosensors with potential applications in cancer research”

  1. Out of curiosity, what is known regarding the glycans expressed by rotifer species and how could this hint at the discovery of potentially new lectins? Could also some of these novel glycans have potential as biotherapeutics and/or tools for glycobiology?

 To the best of my knowledge, no information is available about this subject in Rotifera. Nevertheless, we have previously used a similar approach in other poorly studied lophotrochozoan species, such as Mytilisepta virgata, which expresses asialo-GM1 like glycoconjugates in the mantle and gills (Fujii et al, FEBS J). In that case, the combination between glycobiological and bioinformatics approaches allowed the identification of an asialo GM1 binding lectin in the mantle of this mussel species. In general, although the study of the glycans expressed in the tissues of neglected lophotrochozoan phyla is still quite limited, it appears that each organism displays its own characteristic glycan array, which may also include glycans that are uncommon in other animal groups. This is even more intriguing in the case of rotifers, which are highly divergent from all other extant lophotrochozoan phyla. The expression of these glycans in rotifers could support the identification of novel lectins, which may even belong to previously uncharacterized structural classes.

  1. Keywords: Microbe-Associated Molecular Patterns

Thank you, this has been corrected

  1. Line 52: Damage-Associated Molecular Patterns

Thank you, this has been corrected

  1. Line 86: ‘80s

Thank you, this has been corrected

  1. Table 1: It would look better if the species name were written without cuts (sor- dida), like Adineta ricciae and Brachionus plicatilis are. Also, if it would be possible to add horizontal lines to clearly distinguish visually which species belong to Bdelloidea and to Monogononta.

The table was updated as requested.

  1. Line 138: according to conventions in writing carbohydrates, the “N” of N-acetylglucosamine needs to be in italics.

Thank you, this has been corrected

  1. Line 173: according to conventions in writing carbohydrates, “D” and “L” needs to be smaller. Carbohydrate Research (ISSN 0008-6215) recommends writing these characters as 2 font size smaller than the text.

Thank you, this has been corrected

  1. Line 369: here, SUELs are described as “rhamnose-binding lectins”, yet on line 173, they are also described as “D-gal/L-rha binding SUEL lectin domain”. Would it be possible to adjust this inconstistency, maybe by adding some more information in the results section?

Thank you for pointing this out. The first sentence of the section about this lectin family now reads: “In sea urchins, a group of lectins, characterized by the presence of a D-galactoside/L-rhamnose binding SUEL (acronym for Sea Urchin Egg Lectins) domain, carry out egg-protecting functions”

  1. Line 382: same as comment 9.

Thank you, this has been corrected

Reviewer 3 Report

This is a welcome paper to the literature on diversity of bioactive molecules in lophotrochozoan animals. The central issue addressed in the manuscript is a description and classification of pattern-recognition molecules of lectin nature in six species of phylum Rotifera. This phylogenetically important taxon has so far eluded the attention of researchers looking for novel lectins with unusual and interesting biological properties. This is a big gap in current knowledge, and Gerdol's study suggests ways to close it. This is pioneering work. This study provides a solid basis for further research both on functional characterization and comparative analyses of Rotifera lectins.

The manuscript is clearly written, terminology meets accepted standards, methods are adequate and well described, and references are exhaustive. The presentation of the results and their discussion are scientifically sound, and the author made a good job into placing them in an appropriate context. I see significant citation potential in this manuscript and recommend it for publication in its current state.

Author Response

I would like to thank the reviewer for his/her positive assessment of this work